# Utilizing Protein–Peptide Hybrid Microarray for Time-Resolved Diagnosis and Prognosis of COVID-19

**DOI:** 10.3390/microorganisms11102436

**Published:** 2023-09-28

**Authors:** Peiyan Zheng, Baolin Liao, Jiao Yang, Hu Cheng, Zhangkai J. Cheng, Huimin Huang, Wenting Luo, Yiyue Sun, Qiang Zhu, Yi Deng, Lan Yang, Yuxi Zhou, Wenya Wu, Shanhui Wu, Weiping Cai, Yueping Li, Xiaoneng Mo, Xinghua Tan, Linghua Li, Hongwei Ma, Baoqing Sun

**Affiliations:** 1Department of Clinical Laboratory, National Center for Respiratory Medicine, National Clinical Research Center for Respiratory Disease, State Key Laboratory of Respiratory Disease, Guangzhou Institute of Respiratory Health, The First Affiliated Hospital of Guangzhou Medical University, Guangzhou 510120, China; gdmcslxx@126.com (P.Z.); jasontable@gmail.com (Z.J.C.); huanghuimin311@126.com (H.H.); xveyin@163.com (W.L.); wushanhui2020@126.com (S.W.); 2Guangzhou Institute of Clinical Medicine of Infectious Diseases, Guangzhou Eighth People’s Hospital, Guangzhou Medical University, Guangzhou 510440, China; polinlbl@163.com (B.L.); gz8hcwp@126.com (W.C.); gz8hlypicu@126.com (Y.L.); moxiaoneng@126.com (X.M.); gz8htxh@126.com (X.T.); llheliza@126.com (L.L.); 3Division of Nanobiomedicine, Suzhou Institute of Nano-Tech and Nano-Bionics, Chinese Academy of Sciences, Suzhou 215123, China; jyang2018@sinano.ac.cn (J.Y.); chh22@mail.ustc.edu.cn (H.C.); ustcsyy@163.com (Y.S.); wanx@mail.ustc.edu.cn (Y.D.); yanglbio@foxmail.com (L.Y.); zyx1997@mail.ustc.edu.cn (Y.Z.); wuwenya4925@163.com (W.W.); 4Nano Science and Technology Institute, University of Science and Technology of China, Suzhou 215123, China; 5State Key Laboratory of Respiratory Disease, Guangzhou Institutes of Biomedicine and Health Chinese Academy of Sciences, Guangzhou 510530, China; zhu_qiang@gibh.ac.cn

**Keywords:** receptor binding domain (RBD) probe, sero-IgG dynamic (IsD) events, PPHM assay, serological assays, SARS-CoV-2

## Abstract

The COVID-19 pandemic has highlighted the urgent need for accurate, rapid, and cost-effective diagnostic methods to identify and track the disease. Traditional diagnostic methods, such as PCR and serological assays, have limitations in terms of sensitivity, specificity, and timeliness. To investigate the potential of using protein–peptide hybrid microarray (PPHM) technology to track the dynamic changes of antibodies in the serum of COVID-19 patients and evaluate the prognosis of patients over time. A discovery cohort of 20 patients with COVID-19 was assembled, and PPHM technology was used to track the dynamic changes of antibodies in the serum of these patients. The results were analyzed to classify the patients into different disease severity groups, and to predict the disease progression and prognosis of the patients. PPHM technology was found to be highly effective in detecting the dynamic changes of antibodies in the serum of COVID-19 patients. Four polypeptide antibodies were found to be particularly useful for reflecting the actual status of the patient’s recovery process and for accurately predicting the disease progression and prognosis of the patients. The findings of this study emphasize the multi-dimensional space of peptides to analyze the high-volume signals in the serum samples of COVID-19 patients and monitor the prognosis of patients over time. PPHM technology has the potential to be a powerful tool for tracking the dynamic changes of antibodies in the serum of COVID-19 patients and for improving the diagnosis and prognosis of the disease.

## 1. Introduction

The sudden outbreak of SARS-CoV-2 has been declared a public health emergency of international concern by the World Health Organization [1,2,3,4]. SARS-CoV-2 mainly infects the human lower respiratory tract (lungs) and causes various clinical symptoms such as cough, fever, fatigue, and tachypnea [5,6,7]. According to the different manifestations of pulmonary imaging and clinical symptoms, COVID-19 infections can be classified into four groups (mild, moderate, severe, and critical) [6]. The method of adopting specific strategies to treat patients based on the severity of their condition is currently highly efficient, and most “mild” or “moderate” patients can recover quickly [8]. However, in the case of misclassification of the severity level or insufficient medical care, moderate patients may develop severe or critical conditions, and the risk of death for critical patients increases sharply [6,9]. Thus, a time-resolved diagnosis to accurately estimate the disease status, progression, and prognosis is key to reducing mortality rates of COVID-19 patients.

Lower sensitivity for diagnosis and limited information for prognosis are two well-known shortcomings of traditional receptor-binding domain (RBD)-based serological assays [10]. Although they are mature technologies, RBD-based serological assays typically achieve a sensitivity of approximately 85% [4] owing to the so-called window periods for IgM and IgG production. Recent studies demonstrated that anti-RBD IgG sero-dynamic curves provide limited information on the prognosis, as anti-RBD IgG levels plateau within 14 days post-onset (d.p.o.) and remain at that level even when the severity of the COVID-19 case changes [11].

The humoral immune response can produce functional antibodies (e.g., neutralizing antibodies) to effectively clear the viruses and plays a pivotal role in blocking viral infections [12,13,14]. Previous studies on human immunodeficiency virus, influenza virus, and other viruses have reported that the differences in antibody dynamics during the viral acute infection period in the humoral immune response are linked to differential disease outcomes [15,16,17,18]. Few studies have successfully shown the differences in antibody dynamics against viral proteins in the humoral immune response among different types of COVID-19 patients. SARS-CoV-2 has an abundant number of B-cell linear epitopes [19,20,21,22,23], warranting further studies on the differences in antibody dynamics among different groups of COVID-19 patients from the viewpoint of B-cell linear epitopes.

Upon infection with the SARS-CoV-2 virus, the cellular microenvironment consequently undergoes an acute change [20,24,25]. Significant differences in cytokine levels (e.g., IL-6) and plasma protein levels (e.g., C-reactive protein, CRP) were reported among different groups of COVID-19 patients [26], and both indices were considered potential biomarkers for the diagnosis, progression, and prognosis of COVID-19. Therefore, the identification of biomarkers for diagnostic measures and the development of antigenic targets for vaccines are highly important. Peptide microarrays have been shown to display large numbers of putative target proteins translated into overlapping linear (and cyclic) peptides for multiplexed, high-throughput antibody analysis [9].

In this study, we aim to investigate the opportunities for harnessing B-cell linear epitopes to support COVID-19 diagnosis and prognosis using a discovery cohort and a quarantine cohort, both comprising longitudinal serum samples (i.e., sequential serum samples from a single patient). We introduce a novel diagnostic approach based on protein–peptide hybrid microarray (PPHM) high-throughput screening. This method uses multiple phases of sero-antibody dynamics to capture the changes in the antibody dynamics in serum against different polypeptide epitopes or protein antigens at high resolution. The PPHM platform was used to identify a SARS-CoV-2 epitope containing short peptides (ECSP). Our findings demonstrate that PPHM can provide earlier diagnosis and more accurate prognosis of COVID-19 compared to traditional methods. We also show that PPHM can be used to classify patients into different disease-severity groups and monitor disease progression over time. We hope that our findings will contribute to the ongoing efforts to combat the COVID-19 pandemic and provide a useful tool for clinicians and researchers working in this field.

## 2. Materials and Methods

### 2.1. Ethics Approval

This study was approved by the Medical Ethics Committee of the First Affiliated Hospital of Guangzhou Medical University (ethics approval no. gyfyy-2020-76). All study participants provided written informed consent.

### 2.2. Patients and Serum Sample Collection

This study included two cohorts (discovery and quarantine) and followed the experimental design shown in Appendix A. The discovery cohort (Appendix A) comprised 20 COVID-19 patients (patient #1 to patient #20, a total of 323 serum samples) who were confirmed to have SARS-CoV-2 infection using reverse transcript quantitative polymerase chain reaction (RT-qPCR) and were admitted or transferred to the First Affiliated Hospital of Guangzhou Medical University in February 2020. These patients were initially diagnosed with different disease severity levels based on the classification guidelines of the COVID-19 Diagnosis Program (5th edition) [27]. Respiratory swabs, sputum, and serum samples were collected at different time periods after the symptom onset. Clinical data were retrieved from the medical records of the patients. Among these patients, four with moderate COVID-19 were cured and discharged from the hospital; we labeled these four patients the “moderate-cured” group. Thirteen patients with severe/critical COVID-19 who survived and were discharged from the hospital by the time we started our study (end of June 2020) were labeled the “severe/critical-cured” group. The remaining three patients who were in critical condition and still in the intensive care unit (ICU) at the end of June 2020 owing to other comorbidities were labeled the “critical” group (Appendix A). The gender distribution was different among the patient groups in the discovery cohort, with a higher proportion of males among the critical group, a finding similar to those of previous reports [28,29] (Appendix A). There were no differences in the median age of the patient groups in the discovery cohort (Appendix A).

Among 60 patients in the quarantine cohort (individuals who were diagnosed with COVID-19 during mandatory quarantine upon their entry to China), male patients accounted for 53.3% of the total, and their median age was 45. The majority of the patients had moderate cases of COVID-19 (33/60, 55%). The average d.p.o. for the quarantine cohort was 5.8; this cohort supplied the early-stage samples, which were lacking in the discovery cohort.

We had access to many longitudinal serum samples collected at multiple time points throughout the course of the disease for each patient (Appendix A); thus, we were able to develop an informative serological screening strategy to detect the responsive antibodies against B-cell epitopes over time. As illustrated in Appendix A, we used PPHM technology to perform the serological screening and the array that contains the probes based on whole SARS-CoV-2 proteins (i.e., containing both conformational and linear B-cell epitopes) and the peptides derived from SARS-CoV-2 proteins (i.e., containing linear B-cell epitopes). We were thus able to examine the antibody levels against both types of probes over time during the progression of the disease. Ultimately, we were able to capture informative data trends, which could not have been acquired using other strategies (e.g., solely whole-protein-based serological assays) [30,31].

### 2.3. Peptides and Proteins

By analyzing the amino acid sequence of the SARS-CoV-2 strain (MN908947), 20-mer peptides with an overlap of 10 aa residues, partially covering four structural proteins (S, N, M, E) of SARS-CoV-2, were chemically synthesized by GenScript (Jiangsu, China), and ultimately yielded 136 peptides. We also purchased the RBD protein (GenScript, Jiangsu, China) and N protein (VACURE Biotechnology, Sichuan, China) of SARS-CoV-2 and set them as protein probes in the experiments.

### 2.4. Real-Time PCR Detection of SARS-CoV-2 Infection

Nucleic acid was extracted from the samples collected via nasopharyngeal swabbing following the instructions of a commercial viral RNA extraction kit (DaAn Gene Co., Ltd. of Sun Yat-sen University, Guangzhou, China). A real-time PCR assay kit targeting SARS-CoV-2 ORF1ab and N gene regions was also purchased from DaAn Gene Co., Ltd. Initially confirmed patients or those with negative nucleic acid test results were evaluated by combining two consecutive and consistent test results, with an interval of at least 24 h between the two tests. 

### 2.5. Fabrication of PPHM-1 Microarray

PPHM-1, with the entire panel of 136 peptides and 2 SARS-CoV-2 proteins, was fabricated as described in Appendix A. Briefly, approximately 0.6 NL of each peptide (0.1 mg/ML) or protein (1 Mm) was printed onto the Ipdms substrate membrane using the non-contact printer sciFLEXARRAYER S1 (Capital Bio, Beijing, China) to form a 4 × 4 array (12 arrays with different probes produced for initial serum screening). In each array, three positive controls were printed with human IgG at a concentration of 10 μg/ML, and one negative control was printed with buffer.

### 2.6. Determination of Peptide Composition for PPHM-2 Microarray

Two batches of peptides were enrolled for the fabrication of PPHM-2: ① 27 peptides (from PPHM-1) that specifically interact with the initial serum samples; ② 18 peptides with low detection-signal values for IgG when screened against PPHM-1 by optimizing the following serum-screening conditions: (i) increasing the concentration of serum samples, (ii) increasing the incubation time of serum samples with the microarray, or (iii) increasing both.

The initial screening of four COVID-19 serum samples (Appendix A) was Serum-(Patient #19)-8 d.p.o. (Serum-(Patient ID)-d.p.o.), Serum-(Patient #20)-18 d.p.o., Serum-(Patient #1)-23 d.p.o., and Serum-(Patient #19)-39 d.p.o. Fifty archived anonymous serum samples were enrolled as a negative control, which included 12 SARS-convalescent samples, 5 samples from prostatitis, 6 samples from patients who were infected with EV71, and 27 samples from healthy donors. One randomly selected serum sample from healthy donors was enrolled for initial screening (Appendix A). The selection of the samples was based on a combination of factors designed to screen specific short peptide antibody combinations for the PPHM-2 and give representative antibody responses in SARS-CoV-2

Forty-five peptides (potential ECSPs) and two proteins (Appendix A) were finally selected and printed in a 4 × 4 array using the same approach as that used for producing PPHM-1. The printing concentration was 0.1 mg/ML for each peptide and 1 Mm for each protein. A total of four arrays were produced for COVID-19 cohort screening.

### 2.7. Serum Screening against PPHM Microarrays

Serum was first diluted 100-fold (40-fold for optimized conditions) with serum-dilution buffer (1% bovine serum albumin, 1% casein, 0.5% sucrose, 0.2% polyvinylpyrrolidone, 0.5% Tween20 in 0.01 M phosphate-buffered saline, Ph 7.4). Thereafter, 100 ΜL of the diluted sample was added to each microarray well and incubated for 30 min (or 2 h) on a shaker (500 rpm, 37 °C); the well incubated with serum-dilution buffer only was set as an experimental control. Thereafter, the microarray was rinsed three times with washing buffer (0.01 M PBST) and incubated with 100 ΜL of horseradish peroxidase (HRP) conjugated goat anti-human IgG (ZSGB-BIO, Beijing, China) for another 30 min on a shaker (500 rpm, 37 °C). Human HRP-IgG was diluted 10,000-fold with peroxidase conjugate stabilizer/diluent (Thermo Scientific, Waltham, MA, USA) for experimental use. Finally, 100 ΜL of one-step Ultra TMB-Blotting Solution (Thermo Scientific) was used to detect the informative signal of IgGs against probes using a microarray imager (Suzhou Epitope, Suzhou, China). The data were processed using IBT software, which was also developed by Suzhou Epitope (Suzhou, China). The signal for each dot was calculated using the following equation: Signal dot = Signal readout − Signal background.

### 2.8. Detection of Dynamic Changes in IL-6 and CRP Tests for COVID-19 Patients

Owing to the medical treatment context for each patient, some standard clinical parameters were not tracked throughout the course of the disease. Overall, the COVID-19 patients enrolled in our study had complete clinical records of routine blood examinations and serological assays. The frequency of these examinations and assays was determined by physicians, and the results of the serological assays were associated with the data mainly covering the serum levels of cytokines (IL-2, IL-4, IL-6, IL-10, TNF-α, and IFN-γ) and CRP.

## 3. Results

### 3.1. Identification and Characterization of ECSPs for COVID-19 Diagnosis

#### 3.1.1. Identification of ECSPs Using Protein–Peptide Hybrid Microarray

A two-level PPHM screening strategy was employed in this study [25,32,33]. In the first-level screening, 136 peptides extracted from four structural proteins of SARS-CoV-2 (Appendix A) were used to screen one negative sample and four positive samples. A heat map was generated to exclude non-responsive peptides (Appendix A). Approximately 20% of the immunogen-derived peptides were responsive to the four positive samples (Appendix A). Subsequently, 18 potential ECSPs were identified and used for the second-level screening [25]. This involved 45 peptides (potential ECSPs) and 2 proteins (N protein and RBD), comprising PPHM-2 (Appendix A). PPHM-2 was then used to screen 323 serum samples of 20 COVID-19 patients (discovery cohort) (Appendix A). Thus, IgG levels against both protein and peptide probes were monitored throughout the course of the disease to obtain the IsD curves of the peptide probes.

#### 3.1.2. Characterization of ECSPs Using Serological Assays

At different detection time points, a positive IgG signal value (signal value ≥ 10) against a potential ECSP at one time point was regarded as an ECSP IsD curve. Counts for the ECSP IsD curves in the moderate-cured group were lower than those in the severe/critical-cured or critical group (Appendix A), suggesting weaker humoral immune responses. Unlike the protein IsD curves, which display simple, similar dynamic changes [34], the ECSP IsD curves intertwine in patient #1 (Appendix A). Dissection identified six different types of ECSP IsD curves in patient #1, which provided more detailed information than the protein IsD curves (Appendix A).

The accepted model for the IgG lifecycle, which includes three distinct stages [10] and corresponds to ① rising, ② plateau, and ③ decreasing levels of IgG production (Appendix A), could explain the observed ECSP IsD curves. These curves reflected a single stage of IgG production, the lifecycle of IgG production, or multiple cycle combinations (Appendix A).

The humoral immune system produces IgGs which recognize various antigens [35], including whole proteins and peptides, some of which recognize only whole proteins and conformational epitopes, while others recognize only peptides (e.g., internal linear epitopes, Appendix A). The limited number of interacting epitopes with peptide probes in PPHM diminish compensation of IgG signals from different interacting epitopes, enabling the observation of detailed stages of the epitope-specific IgG lifecycle, i.e., the ESCP IsD curves (Appendix A).

Analysis of the IgG signals against different ECSPs at various time points revealed different-looking curves in the same period and indicated multiple distinct wave phases. Placing three randomly selected ECSP IsD curves together (Appendix A) demonstrated an example of a “distinct wave phase” scenario. Thus, we used the term “multiple phases of antibody sero-dynamics” (MPAD) to refer to the collections of IsD curves (both proteins and ECSPs) for each patient. MPAD was evident in the PPHM screening results of the discovery cohort (Appendix A).

### 3.2. Comparison of ECSP IsD Curves with RBD IsD Curves for COVID-19 Diagnosis

#### 3.2.1. Results of PPHMCOVID-19 Assay

To compare IgG dynamics against different antigens, eight ECSPs and one protein (RBD) were selected from the PPHMCOVID-19 assay for high-efficiency COVID-19 diagnosis (referred to as PPHMCOVID-19 afterward). DMI is the sum of the assigned values of the eight peptide probes and one protein probe (RBD). We proposed four types of results according to antibody development upon SARS-CoV-2 infection:Type #1 is negative (DMI < 2 and anti-RBD IgG negative);Type #2 is positive (DMI ≥ 2 but anti-RBD IgG negative);Type #3 is positive (DMI ≥ 2 and anti-RBD IgG positive);Type #4 is negative (DMI < 2 and anti-RBD IgG positive).

#### 3.2.2. Early Diagnosis by Type #2 Results

Among the 60 patients in the quarantine cohort, 42 were symptomatic and positive for serological assays. These were divided into three groups based on the number of types of PPHMCOVID-19 identified along their IsD curves (Appendix A). Group 1 (14/40, 35%) sequentially showed type #1, type #2 and type #3 results (Appendix A and Figure 1A). This enabled earlier COVID-19 diagnosis as two or more anti-ECSP IgGs entered the seropositive period earlier than the anti-RBD IgG (first identification of type #1 results was 3–14 days post-onset, average 7.5 days; type #2 results 8–20 days post-onset, average 12.8 days; seroconversion lag between anti-ECSP IgGs and anti-RBD IgGs 3–13 days post-onset, average 5.2 days). Group 2 (17/40, 42.5%) showed type #1 results initially, followed by type #3 results (Appendix A and Figure 2B). Type #2 results were absent due to rapid development of anti-RBD IgGs. Group 3 (9/40, 22.5%) showed type #3 results from the first sampling point (Appendix A and Figure 1C). Unusually long incubation periods could explain the missing values for type #2 and type #3 results in this group.

The combination of multiple IsD curves enabled excellent diagnostic performance. Different patients had different anti-ECSP IgG-positive combinations, enabling early diagnosis by type #2 results (Appendix A). Although type #4 result was not obtained due to the limited observation period, some ECSP IsD curves showed a trend of decline (Appendix A). One subgroup of COVID-19 patients showed unusually early sero-reversion, with a frequency of 5% in the discovery and validation cohorts.

#### 3.2.3. PPHMCOVID-19 Compared to PCR and RBD-Based Serological Assays

Out of the 60 patients in quarantine, 42 were symptomatic and 18 asymptomatic COVID-19 patients. Among the 42 symptomatic patients, 28 had multiple positive PCR results (Appendix A). Of these 28, 11 (39.3%) had PCR, PPHMCOVID-19, and RBD-based serological assays sequentially indicating positive results. For example, patient #E5 had seven paired PCR and serum samples 1–44 days post-onset (d.p.o.); three were positive, while the subsequent five negatives were due to viral clearance by 12 d.p.o. Their PPHMCOVID-19 and RBD-based assays had positive results at 12 and 20 d.p.o., respectively, closing the “blind zone” (Appendix A). We plotted the d.p.o.s of the first positive results of these three assays (Appendix A). PCR, PPHMCOVID-19, and RBD-based serological assays had positive results 1–7 (median 4), 3–14 (median 7), and 8–20 d.p.o. (median 12), respectively. Of the 14 patients with one positive PCR result (Appendix A), only three (21.4%) had positive PCR, PPHMCOVID-19, and RBD-based serological assays. Of the 18 asymptomatic patients, only one (patient #E44; Appendix A) had sequential positive results from all three tests. Nine asymptomatic patients had positive PPHMCOVID-19 and RBD-based serological assay results (50%; median 8 d.p.h.), the other nine negative results. Overall, 25 (41.7%) patients had negative PCR from the first day of hospitalization. PPHMCOVID-19 fills the “blind zone” between PCR and RBD-based tests and offers additional clinically actionable information.

#### 3.2.4. ECSP IsD Curves Revealing Differential IgG Dynamics in Humoral Immune Responses among Patient Groups

To assess whether ECSP IsD curves can be used for prognosis, we performed a high-throughput analysis of the whole MPAD data of the discovery cohort. We identified an area where the ECSPs exhibited moderate response rates (~50%) to all serum samples but an extremely low response rate to samples from a certain patient group (Figure 3A, in the yellow box). Except for P-N25, other ECSPs (P-N37–P-N40) revealed varying response rates among the three patient groups (Figure 3B), implying differences in IsD against them.

**Figure 3 microorganisms-11-02436-f003:**
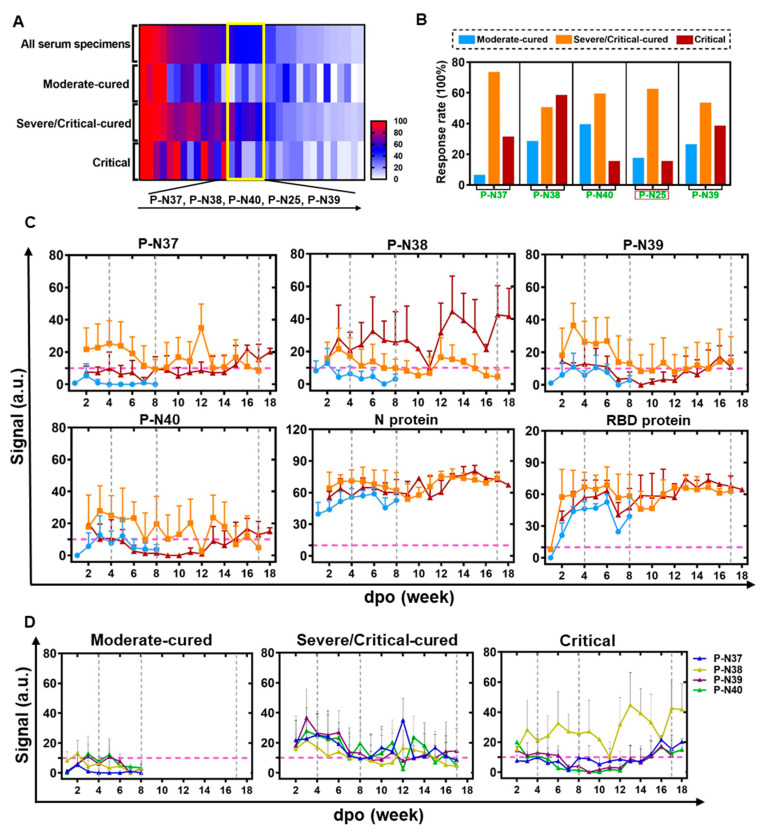
ECSP IsD curves reveal differential IgG dynamics in humoral immune responses among three patient groups. (**A**) A heat map was produced according to the response rates for ECSPs among the three indicated patient groups, based on PPHM-2 screening within the discovery cohort. The yellow box indicates ECSPs which yield moderate response rates (~50%) among patients generally, but which exhibit a low response rate among a single patient group. The ECSPs from the yellow box are detailed at the bottom of the panel. (**B**) Calculation of the response rate for the identified ECSPs in each patient group. (**C**) The IsD curves detected using P-N37~P-N30, N protein, and RBD probes in each of the three patient groups. Data for each time point are shown as the mean + SEM (the average of data for at least three samples). The differential IsD curve patterns were observed in different patient groups during different stages of disease course (defined with the dashed grey line). The color design for the ECSP IsD curves for the patient groups is the same as in Figure 3B. (**D**) The overlaps of ECSP IsD curves of P-N37~P-N40 in the three patient groups. The characteristics of the ECSP IsD curve of P-N38 in the Critical patient group can be used as an index to indicate poor prognosis.

We then determined the IgG signals against ECSPs (P-N37–P-N40) at various time points, resulting in ECSP IsD curves for each patient group (Figure 3C). These curves showed distinct patterns for different patient groups, suggesting differential humoral immune responses (Figure 3C,D). Protein IsD curves among the three groups showed similar trends with almost equivalent IgG levels throughout the disease progression [34,36].

ECSP IsD curves of P-N37–P-N40 were mostly consistent with the patterns depicted in earlier analyses (Appendix A). The only exception here was type ii, which was present in the IsD curves of the N protein (Figure 3C). The ECSP IsD curves of the moderate-cured patients showed a simple curve pattern (Appendix A) four or eight weeks after the symptom onset, reflecting a “short-term” lifecycle of IgG production (Figure 3C,D). The P-N37–P-N40 ECSP IsD curves among the severe/critical-cured patients indicated a long-lasting humoral immune response stronger than that of the moderate-cured patients (Figure 3C).

The ECSP IsD curve patterns of the critical patients were different from the other groups, with varying signals for P-N38 (Figure 3C). This detected response among the critical patients can be used as an indicator of poor prognosis.

ECSP IsD curves of P-N37–P-N40 enable time-resolved monitoring of humoral immune responses of COVID-19 patients with distinct disease statuses. The dynamics of IgG production vary in the early stage of the disease (within four to eight weeks of symptom onset). Further divergence in the IgG dynamics is evident after eight weeks between the severe/critical-cured and critical patient groups. These results suggest that ECSP IsD curves of P-N37–P-N40 can be used to monitor severity, diagnose disease progression, and assess prognosis.

### 3.3. Automatic Severity Classification Based on PPHM Data

Twenty patients were classified into three groups based on their ECSP IsD curves of P-N37–P-N40, which were examined at 0–30, 30–60, and 60–120 d.p.o. (Appendix A). The terms ‘declining’ and ‘rising’ refer to synchronous changes in IgG signals, while ‘irregular’ denotes asynchronous fluctuations. The classification outcome supports the inference that ECSP IsD curves are suitable for predicting COVID-19 prognosis and identifying patients with comorbidities, such as serious inflammatory responses. When combined with PCR testing, ECSP IsD curves improve the time-resolved diagnosis of the disease.

We developed a software program for the automatic processing of PPHM data from longitudinal sera to help clinicians determine disease severity and prognosis. The response trends are automatically determined as declining, oscillating, or persistent. Patients with at least 50% declining response in the first cycle and 100% declining response in the second were classified as “moderate-cured”. Those with 100% persistent or oscillating responses in the third cycle were considered “severe/critical”. Of the discovery cohort of 20 patients, 19 (95%) had their cured status correctly predicted.

### 3.4. ECSP IsD Curves for Predicting COVID-19 Prognosis

Infection-related biomarkers such as IL-6 and CRP are widely used to predict disease progression and prognosis. However, they are known to be sensitive to antibiotics and drugs that are commonly used to treat inflammatory responses, which are frequent comorbidities with infectious diseases [5,37,38,39]. On the other hand, antibody production levels are less influenced by antibiotics [40].

We investigated whether ECSP IsD curves of P-N37–P-N40 provide more clinically accurate and actionable information than IL-6 and CRP as biomarkers for prognosis. Recent studies have indicated that IL-6 and CRP can be used to assess COVID-19 disease progression and prognosis [26]. Generally, a return of IL-6 or CRP to normal levels indicates a good prognosis (Appendix A and Figure 1B). However, irregularities in the P-N37–P-N40 IsD curves can indicate a patient with comorbidities such as serious inflammatory responses (Appendix A). It is evident that the first two patients in Section A show a clear decline in antibody levels before day 60 (D60). However, a similar trend is observed for patient #7 in Group B. Additionally, when examining the kinetics for patient #3 in Group A, there is a modest yet significant increase in antibody levels at D90, a pattern also observed for patient #12 in Group B.

The data of patient #4 revealed the potential for PPHM COVID-19 data to be more informative than IL-6 or CRP data. This patient was placed in the moderate-cured patient group, and test results indicated viral presence at 7–17 d.p.o./19–42 d.p.o. and viral clearance at 17–19 d.p.o./after 47 d.p.o. (Figure 2C). Two distinct P-N37–P-N40 IsD curves reflected the virus status change. Anti-ECSP IgG production after 19 d.p.o. was characteristic of critical patient group patterns (Figure 3D). This oscillating biomarker data explained the patient’s initial misclassification as a critical patient, and also captured the viral clearance process that IL-6 trends may have missed (Figure 2C). These results suggest caution should be taken when using IL-6 levels as biomarkers for COVID-19 treatment.

### 3.5. Differentiating COVID-19 Severity Levels through ECSP IsD Curves

Three checking cycles for PMI model were defined as “0–30 dpo” (first checking cycle), “30–60 dpo” (second checking cycle), and “60–120 dpo” (third checking cycle), respectively (Figure 4). The disease severities of the confirmed COVID-19 patients can be initially distinguished according to the ECSP IsD curves of P-N37~P-N40 during the first checking cycle. The patients with P-N37~P-N40 IsD curves similar to Image ① can be confidently diagnosed as moderate patients who should recover soon with standard treatments; these patients will not require further assessment in the second or third checking cycles. In contrast, patients with P-N37~P-N40 IsD curves similar to Image ② or ③ should be understood as severe or critical patients; these patients will require further assessment during the second checking cycle. If, during the second checking cycle, the ECSP IsDs appear similar to Image ④, these patients are recovering well and will likely become moderate patients. One of two alternatives at this point is patients with ECSP IsD curves that appear similar to Image ⑤; such patients should be given continuous examination throughout the third checking cycle, and if their ECSP IsD curves (or at least three ECSP IsD curves) appear similar to Image ⑤/⑦ during the third checking cycle, such patients are recovering from the disease with a curable outcome. The second alternative is when the second checking cycle ECSP IsD curves appear similar to Image ⑥; these critical patients show irregular anti-(P-N37~P-N40) IgG production that may reflect serious comorbidities (inflammatory responses), and they should be treated with intensive care as soon as possible. Such patients may have third checking cycle ECSP IsD curves that appear similar to Image ⑧, which indicates a very poor prognosis.

## 4. Discussion

For traditional whole-protein-based serological assays, the detection of an antibody in a single serum sample may reflect either a previous infection/vaccination or an ongoing infection. To overcome this ambiguity, the currently promulgated practice guideline to confirm an ongoing infection is to require evidence for either (i) a seroconversion or (ii) a greater than fourfold increase in the “specific” IgG level. Our analysis of IsD curves emphasized that meeting these two requirements is exceedingly difficult in practice [10].

We previously conducted a PPHM-based study, which detected a short-lived characteristic for anti-ECSP IgGs in peste des petits ruminants virus (PPRV) vaccination [25]. We successfully used this feature to differentiate between infected and vaccinated animals for a live attenuated PPR vaccine. Specifically, by 40 days post-vaccination (d.p.v.), anti-ECSP IgGs were no longer detectable, and the detection of anti-(F protein) IgGs indicated the effectiveness of the vaccine. If anti-ECSP IgGs were detected after 40 d.p.v.—regardless of the level of anti-(F protein) IgGs—then we could infer that PPRV infection must have occurred. In the present study, we found the same short-lived characteristic for anti-ECSP IgGs in SARS-CoV-2 infection, and hypothesized that a positive anti-ECSP IgG signal from PPHMCOVID-19 can confirm ongoing infection based on a single serum sample.

Our analyses of longitudinal serum samples from the two cohorts supported the following conclusions: (1) anti-ECSP IgGs are short-lived relative to anti-RBD IgGs; (2) ECSP IsD curves from PPHM data support an earlier diagnosis than traditional whole-protein based serological assays; and (3) PPHMCOVID-19 can cover the blind zone of COVID-19 diagnosis created by PCR- and RBD-based chemiluminescent immunoassay (CLIA) assays. The practical utility of these conclusions is evident considering that latitudinal serum samples are typically the only available samples in common clinical practice. In the study conducted in parallel using a large latitudinal sera cohort, PPHMCOVID-19 was applied to latitudinal serum samples and achieved earlier diagnosis and higher sensitivity than traditional whole-protein-based serological assays. Thus, the longitudinal serum analysis suggests that some anti-ECSP IgGs have earlier seroconversion than anti-RBD IgG, and thus PPHM has the capacity for an earlier diagnosis than traditional whole-protein-based serological assays.

Current strategies for predicting COVID-19 disease progression and prognosis are based on the diagnostic results obtained from a few time points at a very early stage of the disease course [26,41]. In this study, we used the information that MPAD data from PPHM testing can provide to accurately predict the disease status and progression in real time. This time-resolved diagnosis and prognosis can support improved clinical care. Continuous monitoring can prevent the misevaluation of the disease progression or final disease outcome. A cytokine storm is known to be a manifestation of the severity of virus-linked diseases, and can occur in COVID-19 patients [42,43]. Both antibiotics and hormone drugs are commonly used to manage the inflammatory responses of associated comorbidities in COVID-19 patients. Although these medications can quickly return the levels of infection-related biomarkers (e.g., IL-6 and CRP) to normal levels, patients treated with these drugs are at risk of virus recurrence (such as Patient # 4) [44,45,46]. The lifecycle of antibody production is less sensitive to antibiotics and hormone drugs than the monitoring of inflammatory responses [37,39]. These results indicate how IL-6 trends can be affected by the administration of antibiotics and hormonal drugs, and that caution should be exercised when using IL-6 levels as biomarkers for COVID-19 treatment. While several treatments have emerged for COVID-19, novel cell-based approaches have shown significant potential in managing the disease [47]. Accordingly, using ECSP IsD curves to monitor antibody production is more reliable and safer for prognosis and clinical applications.

The main limitation of this study is the lack of a validation group for the prognosis test. That is, the prognosis potential of P-N37–P-N40 was only explored in a small discovery cohort. Although we did assemble a training group (Appendix A), retrospective studies that included samples representing the highly informative “third checking cycle” (Figure 3) for these patients were not available due to the limited sampling duration. An ideal validation group for assessing the prognosis performance of P-N37–P-N40 would comprise data from a larger cohort that specifically includes serum samples taken during the third checking cycle.

Despite this limitation, the context of the ongoing COVID-19 pandemic supports our reporting of the potential of using P-N37–P-N40 IsD curves for prognosis. This study highlights the unique application potential of MPAD based on PPHM data and will be of interest to researchers and clinicians who work in this field. Using the data provided by MPAD, we are planning to develop an algorithm for mining neutralization- and indicator-related ECSPs, involving heuristic inputs as well as machine learning methods to accommodate the full scope of available information. Equally important are carefully planned clinical cohort studies that include systematic serological sampling across the full disease course. ECSP IgG sero dynamic (IsD) curves enable substantially earlier COVID-19 diagnosis than RBD IsD curves (e.g., 6 days earlier). We performed a retrospective study to analyze the PPHM data collected from the discovery cohort: (i) at different time points after symptom onset, (ii) in multiple individual patients, with (iii) different disease severities, and (iv) differential disease outcomes. Four ECSPs showed prognosis potential, and the severity classification of COVID-19 patients was completed using an automated software program that supports the automatic processing of PPHM data from longitudinal sera. Even if the lack of data on immunocompromised people may limit the generalizability of our results, we believe that this methodology could be adapted to study these special populations and could offer valuable insights into their antibody responses. The noteworthy implication of this study lies in its application of machine learning to accelerate the detection and classification computational processes for the large datasets generated by the microarray. Therefore, a deep collaboration with data scientists to explore how to integrate machine learning into the PPHM platform will be a direction for the future development of this technology.

## 5. Conclusions

In conclusion, our study demonstrates that PPHM provides a more comprehensive and high-resolution view of sero-antibody dynamics in COVID-19 patients than traditional single-protein methods. While PCR remains the diagnostic gold standard, PPHM contributes valuable supplementary data on the host’s immune response. The method is particularly useful in monitoring post-infection immunity in patients at risk of reinfection, evaluating disease progression in hospitalized patients, and assessing vaccine efficacy. Our preliminary findings also indicate that the early convalescent phase (2–3 weeks post symptom onset) is the optimal time for blood-sample collection to obtain a robust antibody profile. Furthermore, the validation of our proposed prognostic algorithm in a larger cohort with longitudinal sera samples from different stages of the COVID-19 disease course is warranted to further establish its clinical utility and accuracy.

## Figures and Tables

**Figure 1 microorganisms-11-02436-f001:**
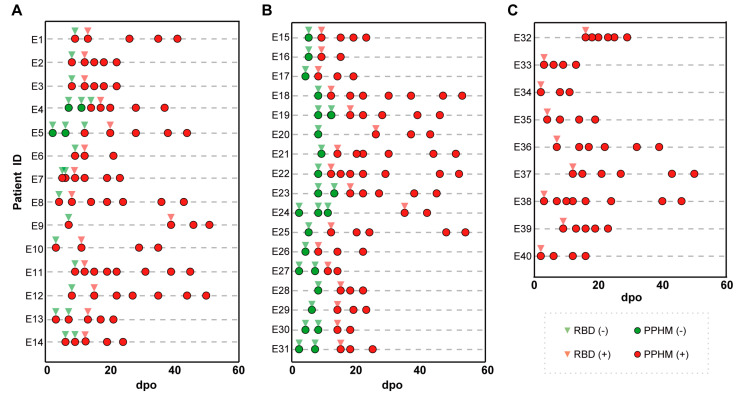
The representative patient information of the three groups, as well as the detection time and results of PPHM COVID−19. (**A**) A total of 14 patients of group 1 obtained an earlier diagnosis because of a situation in which two or more anti-ECSP IgGs entered the sero-positive period earlier than the anti-RBD IgG; (**B**) 17 patients were categorized into Group 2; (**C**) 9 patients were categorized into Group 3.

**Figure 2 microorganisms-11-02436-f002:**
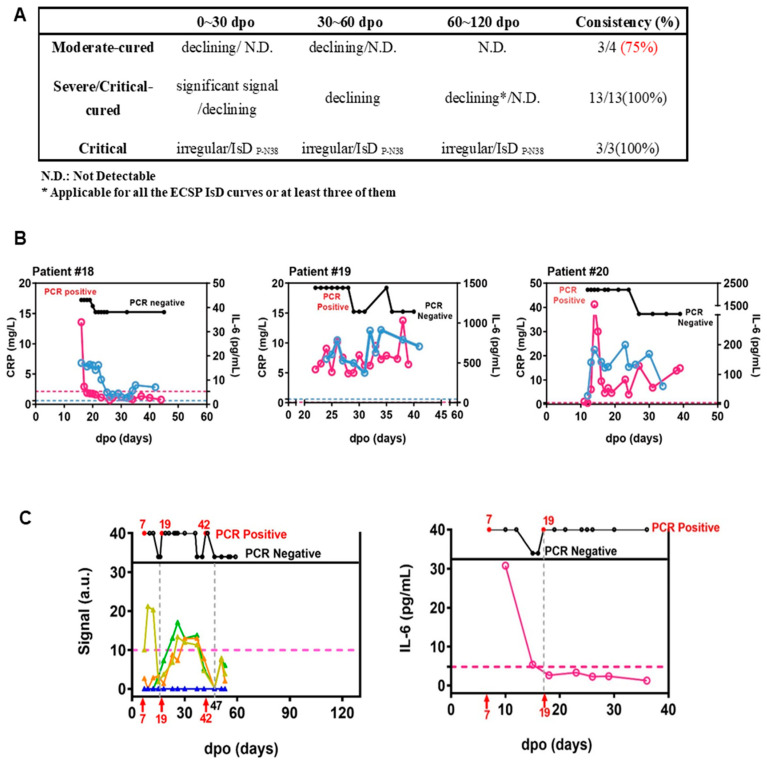
ECSP IsD curves outperformed infection-related biomarkers in predicting COVID-19 prognosis. (**A**) The predictive ability of ECSP IsD curves in patients with different severity levels at different periods. (**B**) The dynamic changes in both IL-6 levels (purple) and CRP levels (blue) for three COVID-19 patients from the Critical patient group. The threshold for normal levels of IL-6 and CRP are presented with dashed lines of the corresponding color. High levels of IL-6 and CRP indicate poor disease prognosis. (**C**) The ECSP IsD curves of P-N37~P-N40 (bottom) and the dynamic changes of PCR results detected at different time points (top) for Patient #4 (left). the color design used for these four ECSP IsD curves is the same as in Figure 3D, and the threshold for a valid IgG signal is presented as a dashed line. The dynamic changes in the IL-6 level for Patient #4 (right); a normal IL-6 level is presented as a dashed line.

**Figure 4 microorganisms-11-02436-f004:**
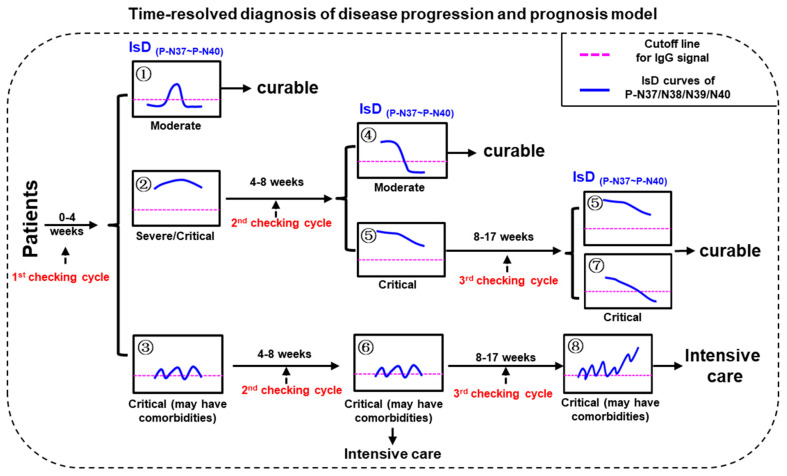
The model for time-resolved diagnosis of COVID-19 disease progression and prognosis (PPHM-MPAD-IsD “PMI” model).

## Data Availability

The data that support the findings of this study are partially available in the Appendix A of this article. The full dataset generated and/or analyzed during the current study are not publicly available due to ethical considerations, but are available from the corresponding author upon reasonable request.

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
