# Peer review of "Utilizing Protein–Peptide Hybrid Microarray for Time-Resolved Diagnosis and Prognosis of COVID-19"

_microorganisms, 2023, doi:10.3390/microorganisms11102436_

Round 1

Reviewer 1 Report

In this paper, the authors describe the use of protein-peptide hybrid microarray (PPHM) to study the dynamic changes of antibodies against SARS-CoV-2 in the serum of COVID-19 patients. They use 2 cohorts of patients, a "discovery" cohort of 20 patients, and a "Quarantine cohort" of 60 patients. They have identified 4 peptides that they consider to be able to discriminate the severity stages of patients and they conclude that PPHM technology could improve the diagnosis and prognosis of COVID-19 patients.

Major comments:

- A more precise prescription of the patients, and of the treatment they received could be of interest, since we ca hypothesize that different treatments (corticosteroids, for example) could modify the immune response. Particularly, in both cohorts, did some patients have immunosuppressive treatments (for organ transplantation, for example, or immunosuppressed by HIV). If not, is it a possible limitation to the  results indicated here ? Can this methodology be useful for such patients ?

- Fig 7 S is very interesting but do not illustrate the discriminant capacity of the study of the profiles of the patients; It is clear thta the 2 first patients in A section have a clear decrease of the antibodies, before day 60 (D60). However, a smilar profile is observed for patient #7 in group B. And if you analyse the kinetic of the patient #3 in the groupe A, there is a modest but real increase of antibodies at D90, a similar profile that is observed for patient #12 in the group B. The reviewer consider that this figure is very important and should not be consider as "Supplementary date", but should be present in the text.  And the varaiations according to the different patients as mentionned above discussed.

- In general, the text is sometimes a little complex to understand, and the distribution between text figures and additional data should be reviewed. For example, Figs 1 and 2 and 3 could be supplementary material, while Figures S4 and S7 could be put in the main text. There are many details in the text which are not really useful, we do not see why the authors mention the different PCRs carried out (sputum, urine), the sensitivity of which is often lower than the nasopharyngeal samples. How did this help them validate their test? 

- In contrast some explanations should be given concerning the choice of serum for the initial screening for the fabraication of PPHM-2, in particular it is written "12 SARS convalsecent samples" (pge 4, line 185);. The authors should describe how they choose these ssamples, and why.

- As a general rule, COVID-19 diagnosis by PCR no longer poses many difficulties at present. It seems, from the data presented here, that the PPHM matode still requires kinetics; The authors should add a paragraph in the discussion indicating to which patients this test should be indicated, and when to take the blood sample, because all this has a financial cost.

References 5 to 7 are in an unusual format.

Minor comments:

- page 6, line 276 : in the quarantine cohort, 40 patints were indicated as "symptomatic", whereas 42 are indicated in the same page, line 296.

- In the Fig S&, from patients 6, "19" is written for all the columns, whereas the height of the columns suggests that these are different values.

Reviewer 2 Report

The COVID-19 pandemic has underscored the pressing demand for rapid, precise, and cost-effective diagnostic techniques, given the limitations of traditional methods like PCR and serological assays. This research focused on exploring the application of protein-peptide hybrid microarrays (PPHM) to monitor changes in antibodies among COVID-19 patients and predict disease severity and prognosis. The study involved a cohort of 20 COVID-19 patients, demonstrating the efficacy of PPHM in tracking antibody dynamics. Specifically, four distinct polypeptide antibodies emerged as valuable for assessing patient recovery and forecasting disease progression.

Regarding the sample size, it raises the question of whether this cohort of 20 patients is sufficiently robust to yield conclusive results. Additionally, the study's focus on humoral immunity prompts consideration of whether efforts were made to characterize detected IgGs, categorizing them as IgG1, IgG2a, IgG2b, or IgG3. Such classification could enhance our understanding of Th1 and Th2 immune responses. Furthermore, while the study primarily examined humoral immunity, it's worth exploring how this platform could potentially be leveraged to monitor adaptive immune responses, as this facet of the immune system may also influence COVID-19 prognosis.

Examining the limitations of the current platform is crucial. Incorporating machine learning tools into the algorithm could be an intriguing avenue to explore, potentially enhancing both sensitivity and specificity. This addition could fortify the platform's performance and its ability to distinguish subtleties in the antibody profiles, contributing to more accurate disease monitoring and prognosis prediction.

Reviewer 3 Report

Peiyan Zheng and co-authors present a quality and well-written manuscript describing utilizing protein-peptide hybrid microarray for time-resolved diagnosis and prognosis of COVID-19.

Authors investigated the potential of using protein-peptide hybrid microarray (PPHM) technology to track the dynamic changes of antibodies in the serum of COVID-19 patients and evaluate the prognosis of patients over time.A discovery cohort of 20 patients with COVID-19 was assembled, and PPHM technology was used to track the dynamic changes of antibodies in the serum of these patients. The results were analyzed to classify the patients into different disease severity groups, and to predict the disease progression and prognosis of the patients.PPHM technology was found to be highly effective in detecting the dynamic changes of antibodies in the serum of COVID-19 patients. Four polypeptide antibodies were found to be particularly useful for reflecting the actual status of the patient's recovery process and for accurately predicting the disease progression and prognosis of the patients. 

Authors demonstrate that PPHM can provide earlier diagnosis and more accurate prognosis of COVID-19 compared to traditional methods. Authors also show that PPHM can be used to classify patients into different disease severity groups and monitor disease progression over time.  The findings of this study emphasize the multi-dimensional space of peptides to analyze the high-volume signals in the serum samples of COVID-19 patients and monitor the prognosis of patients over time. PPHM technology has the potential to be a powerful tool for tracking the dynamic changes of antibodies in the serum of COVID-19 patients and for improving the diagnosis and prognosis of the disease.

Finally, authors conclude that the findings of this study emphasize the multi-dimensional space of peptides to analyze the high-volume signals in the serum samples of COVID-19 patients and monitor the prognosis of patients over time. Authors hope that their findings will contribute to the ongoing efforts to combat the COVID-19 pandemic and provide a useful tool for clinicians and researchers working in this field.

==============================

Other comments:

1) Please check for typos throughout the manuscript.

2) With regards to treatment of COVID-19 – authors are kindly encouraged to cite the following article that describes novel cell-based approaches to treat COVID-19. DOI: 10.3390/biomedicines9010059

Round 2

Reviewer 1 Report

Dear authors,

Thank you very much for your detailed answers. I have only one proposition concerning the 2 sentences in the discussion p25, line 745. I suggest combining the two sentences into one: "Even if the lack of data on immunocompromised people may limit the generalizability of our results, we believe that this methodology...."

Author Response

Dear Reviewer,

Thank you for your feedback and suggestion regarding the discussion on page 25, line 745. We appreciate your careful review of our manuscript.

We agree with your suggestion to combine the two sentences to improve clarity and conciseness. The revised sentence will read as follows: "Even if the lack of data on immunocompromised people may limit the generalizability of our results, we believe that this methodology...."

We have made this change in the manuscript accordingly.

Once again, thank you for your valuable input. If you have any further comments or suggestions, please do not hesitate to share them.

Sincerely, 

Baoqing Sun

Reviewer 2 Report

The study has been revised and is now acceptable for publication.

Author Response

Dear Reviewer,

Thank you very much for your positive feedback and for finding our revisions acceptable for publication. We greatly appreciate your guidance and insights throughout the review process.

We are excited about the opportunity to contribute our research to [Journal Name], and we will promptly proceed with the necessary steps for publication.

Once again, thank you for your time and valuable input.

Sincerely,

Baoqing Sun